

# Interpretation of Particle Number Size Distributions Measured across an Urban Area during the FASTER Campaign

**Roy M. Harrison[1][*][†], David C.S. Beddows[1]**
**Mohammed S. Alam[1], Ajit Singh[1], James Brean[1],**
**Ruixin Xu[1], Simone Kotthaus[2] and Sue Grimmond[2]**

**[1]Division of Environmental Health and Risk Management,**
**School of Geography, Earth and Environmental Sciences**
**University of Birmingham**
**Edgbaston, Birmingham B15 2TT**
**United Kingdom**

**[2]Department of Meteorology**
**University of Reading, Reading RG6 6BB**
**United Kingdom**

[*] To whom correspondence should be addressed.
Tele: +44 121 414 3494; Fax: +44 121 414 3709; Email: r.m.harrison@bham.ac.uk

[†]Also at: Department of Environmental Sciences / Center of Excellence in Environmental Studies, King Abdulaziz University, PO Box 80203, Jeddah, 21589, Saudi Arabia



**ABSTRACT**
Particle number size distributions have been measured simultaneously by Scanning Mobility
Particle Sizers (SMPS) at five sites in Central London for a one month campaign in January –
February 2017.  These measurements were accompanied by condensation particle counters (CPC)
to measure total particle number count at four of the sites and aethalometers measuring Black
Carbon (BC) at five sites.  The spatial distribution and inter-relationships of the particle size
distribution and SMPS total number counts with CPC total number counts and Black Carbon
measurements have been analysed in detail as well as variations in the size distributions.  One site
(Marylebone Road) was in a heavily-trafficked street canyon, one site (Westminster University)
was on a rooftop adjacent to the Marylebone Road sampler, a further sampler was located at
Regent's University within a major park to the north of Marylebone Road.  A fourth sampler was
located nearby at 160 m above ground level on the BT tower and a fifth sampler was located 4 km
to the west of the main sampling region at North Kensington.  Consistent with earlier studies it was
found that the mode in the size distribution had shifted to smaller sizes at the Regent's University
(park) site, the mean particle shrinkage rate being 0.04 nm s$^{-1}$ with slightly lower values at low wind
speeds and some larger values at higher wind speeds.  There was evidence of complete evaporation
of the semi-volatile nucleation mode under certain conditions at the elevated BT Tower site.
Whereas SMPS total count and Black Carbon showed typical traffic-dominated diurnal profiles, the
CPC count data typically peaked during nighttime as did CPC/SMPS and CPC/BC ratios.  This is
thought to be due to the presence of high concentrations of small particles (2.5 – 15 nm diameter)
probably arising from condensational growth from traffic emissions during the cooler nighttime
conditions.  Such behaviour was most marked at the Regent's University and Westminster
University sites and less so at Marylebone Road, while at the elevated BT Tower site the ratio of
particle number (CPC) to Black Carbon peaked during the morning rush hour and not at nighttime,
unlike the other sites.  An elevation in nucleation mode particles associated with winds from the



West and WSW sector was concluded to result from emissions from London Heathrow Airport,
despite a distance of 22 km from the Central London sites.



## 1.    INTRODUCTION

The adverse health consequences of air polluted by particulate matter are now well recognised

(WHO, 2006).  While the main focus has been on the public health impact of exposure to fine

particulate matter measured by mass ($PM_{2.5}$), there has also been concern over the possible

contribution of ultrafine particles of less than 100 nm diameter to adverse health outcomes.  While

such particles contribute little to the total mass of particles in the atmosphere, they dominate particle

number (Harrison et al., 2000) and authoritative reviews have concluded that although evidence is

currently highly incomplete, they may contribute to the toxic hazard associated with ambient

particulate matter (HEI, 2013; WHO, 2013).  There have also been suggestions that particle surface

area plays a major role in health impacts and this resides largely in the accumulation mode which is

typically centred around 100-200 nm diameter (Harrison et al., 2000).  Consequently, there is a

strong interest from a health perspective in sub-micrometre particles and there are many reports of

their concentrations and size distributions within the atmosphere.

In addition to concerns over human health, there are other reasons for the study of the size

distribution of airborne particles.  Not only does this strongly influence their location and efficiency

of deposition in the human lung, the particle size distribution can also be a strong indicator of

particle source, with there being some clear differences between the modal diameter of particles

arising from different sources (Vu et al., 2015a).  The clearest distinction is between particles

arising from combustion and other high temperature sources, which tend to be predominantly very

small, and particles generated by attrition processes which are typically far more coarse.  However,

even within the particles generated from combustion and other high temperature sources, there may

well be different modal diameters associated with different sources or even multiple modes

associated with an individual source (Vu et al., 2015a).  For example, exhaust emissions from diesel

engines typically comprise both a nucleation mode and an overlapping Aitken mode, reflecting in

the former case particles comprised mainly of condensed lubricating oil formed after the





combustion process, and in the latter case, solid carbonaceous particles formed within the
combustion process (Shi and Harrison, 1999; Alam et al., 2016).
After their emission, particle size distributions are also liable to change through dynamic processes.
These include evaporation which causes particles to shrink without changing the overall number,
condensation which causes particles to grow without a change in total number, coagulation which
also causes growth but reduces the total particle number, and deposition which causes a reduction in
number and is a strong function of the particle size.
Within this study, particle number size distributions were measured simultaneously by electrical
mobility spectrometers at five separate sites across London and the size distributions are compared
with a view to gaining a better understanding of the sources and processes affecting particles in the
urban atmosphere.
**2.      EXPERIMENTAL**
Data were collected from 27 January 2017 to 16 February 2017 as part of the second campaign of
the FASTER project.  Data recovery was high at all sites except Westminster University, where
good SMPS data were collected on only three days, January 30 and 31 and February 1, 2017.
**2.1     Sampling Sites**
Data were collected at five sampling sites in total, three of which were established specifically for
the FASTER campaign, Westminster University, Regent's University and BT Tower.  The other
two sites (London Marylebone Road and London North Kensington) collect data as part of the
national Automatic Urban and Rural Network.  The site locations (seen in Figure 1) and
characteristics are as follows:





• *Marylebone Road*.  Air sampling equipment is housed in a large kerbside cabin on the sidewalk

of a busy central London street canyon with an inlet approximately 2.5 m above ground-level

(agl).  The adjacent six-lane highway carries around 80,000 vehicles per day.  The highway is

relatively straight and runs almost due east-west (angle 80º from north).  The buildings on

either side of the highway are around six storeys in height giving a street canyon aspect ratio of

approximately 1:1.

• *Westminster University*.  Air sampling instruments were located on the roof of the Westminster

University building, almost directly above the Marylebone Road air sampling site on the

southern side of the street.  The instruments were housed in a temporary enclosure located

approximately 26 m above street level and 4.5 m from the front edge of the roof where it

overlooks the road, and with an inlet 1.5 above the roof.

• *Regent's University*.  A temporary enclosure for the instruments was located on the roof of

Regent's University which is an isolated building within Regent's Park due north (i.e. 360°) of

the Marylebone Road and Westminster University sites.  The only highway lying between

Marylebone Road and the Regent's College site is a lightly trafficked road within Regent's

Park. The distance between the Westminster University and Regent's University sites is

estimated at 380 m.  The instruments were located 16 m agl and 1 m from the edge of the roof.

• *London North Kensington*.  Instruments were sited in a permanent cabin located within the

grounds of a high school in a lightly trafficked suburban area of central London, with an inlet

approximately 2.5 m agl.  The air pollution climate at this site, often taken as representative of

the background air quality within central London, has been characterised in detail by Bigi and

Harrison (2010).

• *BT Tower*.  Instruments were sited on level T35 at approximately 160 m agl on a narrow tower

which rises well above the surrounding buildings on a quietly trafficked street approximately

380 m to the south of Marylebone Road.  The site was used extensively in the REPARTEE

experiment (Harrison et al., 2012a).



## 2.2 Sampling Instruments

The instruments (Table 1) were operated according to Wiedensohler et al. (2012) guidelines and

calibrated and intercompared both before and after the sampling campaign. Small correction factors

(< 5%) were applied to CPC (condensation particle counter) data as a result of the intercomparison.

SMPS (scanning mobility particle sizers) data were analysed using the AIM9 and AIM10 software

provided by TSI as appropriate to the instrument. The national network sites (Marylebone Road

and North Kensington) are fitted with diffusion dryers according to EUSAAR/ACTRIS protocols

(Wiedensohler et al., 2012), but the other sites were not. The particle size ranges measured were

14.9-615.3 nm at Westminster University, Regent's University and BT Tower, 16.55-604.3 nm at

Marylebone Road and North Kensington, and a further system with a short DMA (differential

mobility analyses) gave 4.96-145.9 nm at Regent's University.

It was not possible to use identical SMPS systems at each site. The variants used are shown in

Table 1. We expect little difference between the long column classifiers (TSI 3081) used at all sites

but with different platforms (TSI 3080 and TSI 3082) and CPCs (TSI 3775 and 3776). Differences

are expected to be minimal as platform-specific software was used to invert the data and both the

CPC are butanol-based, with only slightly different lower cut-points which were well outside of the

range of measured particles. At the Regent's University site, both a long DMA (3081) and short

column DMA (3085) were utilised and the data were merged to give a single continuous size

distribution from 6 nm to 650 nm. A possible cause of divergence is the fact that two of the sites

(Marylebone Road and North Kensington) used diffusion dryers according to the EUSAAR/

ACTRIS Protocol. The dryers were tested when installed and showed very low particle losses (less

than 5%) and no significant change to particle size distributions (NPL, 2010). The dryer may,

however, affect the particle size distribution due to the hygroscopicity of certain kinds of particles.

Vu et al. (2015b) reviewed hygroscopic growth factors for submicron aerosols from different

sources. Their data are difficult to extrapolate to this study as measurements of hygroscopic growth



are typically made at very high relative humidities, normally around 90%. Even at 99.5% relative
humidity, the growth of particles of less than 100 nm sampled from the atmosphere is relatively low
(Vu et al., 2015b). Consequently, a reduction in humidity from 88% typical of the campaign to the
values of 30-40% achieved in the dryer would be expected to have only a small effect on particle
sizes especially as fresh traffic-generated particles which comprise a large proportion of the sub-
micrometre particulate matter in the urban atmosphere are hydrophobic and therefore undergo zero
or very limited growth in humid atmospheres.

**2.3      Weather Conditions During the Campaign**
Wind speed and direction data were taken from Heathrow Airport to the west of London to reflect
the synoptic flow minimally affected by local building effects. At the start of the campaign (27
January 2017) the wind direction was easterly and moved to southerly by January 29th, briefly
passing through northerly before returning to a southerly circulation between January 31 and
February 3rd. During this time, wind speeds were typically around 4 m s$^{-1}$ and temperatures mild
for the time of the year (mostly 6-10 ℃). From February 4th to 8th there was a period of lower
wind speeds (1-4 m s$^{-1}$) with variable wind directions and low nocturnal minima temperatures
(down to 1℃). From Feburary 8 – 12$^{th}$, a period of northerly winds (speeds of 3-5 m s$^{-1}$) and lower
temperatures (1-3℃) without appreciable diurnal variation occurred. After February 12th, the
winds came from the east moving to south-westerly by February 17$^{th}$, with wind speeds variable
(between 0 and 6 m s$^{-1}$) and temperatures steadily rising to daily maxima of 12℃.

The mixed layer heights (MLH) were determined from Vaisala CL31 ceilometer data collected at
the Marylebone Road site (Figure 1, Table 1). The observed 15 s (10 m gates) aerosol attenuated
backscatter profiles were pre-processed (Kotthaus et al., 2016) prior to using the CABAM
algorithm (Kotthaus and Grimmond, 2018) to determine 15 min intervals MLH. The multiple
aerosol layers (e.g. nocturnal residual layers) in the atmosphere are detected (Kotthaus and





Grimmond, 2018;  Kotthaus et al., 2018). Here the lowest detected layer is analysed. At times the
MLH cannot be detected (e.g. during rain or very weak gradients in attenuated backscatter), but a
residual layer might still be indicated. The ceilometer detects periods of precipitation, including
events that may not be recorded by ground-based stations (e.g. insufficient to trigger a tipping
bucket rain-gauge).

During the campaign the observed MLH varied from a daily minimum of 45 m agl to a daily
maximum of 1312 m agl with an overall 15 min average (median) of 421 (382) m agl. The daily
average (median) maximum MLH was 777 (695) and minimum was 194 (197) m agl. The daily
range and the amount of data available per day are shown in Figure S10.

**2.4      Modal Analysis of Size Distributions**
Modes were fitted to the 15 min data obtained at Marylebone Road, Regent's and Westminster
Universities using curve fitting and data analysis software "Fityk (version 1.3.1)" developed by
Wojdyr (2010).  In the present analysis, a standard peak function (equation 1) was used to
disaggregate the size distributions into lognormal modes:

$$P_i = A_i \cdot exp\left[-\left(\frac{\ln(D/c_i)}{W_i}\right)^2\right] \tag{1}$$

By fitting linear a combination of n peaks ($P_1 + P_2 + \ldots + P_i + \ldots + P_n$) to the number size
distributions, the following information was calculated: 1) amplitude $A_i$ and location of dN/dlogD at
the mode of the distribution $c_i$, 2) area under the curve (nm cm$^{-3}$), and 3) width of the lognormal
curve $W_i$.






## 3. RESULTS AND DISCUSSION

### 3.1 Particle Size Distributions

A time series of total particle number concentrations from the SMPS instruments appears in Figure 2. A strong diurnal variation is seen at all sites and is exemplified by the average daily variation shown in Figure 3.

The data stratified by LHR wind direction (Figure 4) were used to perform the modal analysis. The log normal modes fit to the size distribution were used to provide insights into the separate modes contributing to a measured size distribution. Although most measurements could be fit with three separate modes some distributions were best fit with only two modes. An example of a three mode fit of a size distribution from North Kensington appears in the data for the 270° wind sector at this site (Figure 5). It may be seen that using three modes gives a very good overall fit to the data. The details of the modes fitted and their relative magnitude and breadth appear in Table S1.

The Marylebone Road sampling site is located in a heavily trafficked (approx. 80,000 vehicles per day) street canyon. The canyon is aligned almost east-west and the sampling site is at kerbside on the southern side of the street. The canyon has a height to width ratio of ~1 consequently we expect skimming flow when flow is perpendicular, with one or more vortices established in the canyon (Oke et al. 2017). When there is one vortex, the sampler is exposed to freshly emitted traffic contaminants when the wind above the canyon is from the south (Figure 6). Particle number concentration on Marylebone Road is highest for the 225° and 270° wind sectors (Figure 4a) when traffic-generated pollutants are carried efficiently to the sampler. When winds have a northerly component such as those for 0° and 45° in Figure 4a, the air reaching the sampler is typical of background air from north London and peak concentrations fall by a substantial margin. The particle size data from Marylebone Road (Table S1) show no strong effect of wind direction on the modal diameter for the first fitted mode in the distribution. The average diameter for the 180 and



225º wind sectors are 21.4 nm while for the 0 and 45º sectors they are 22.9 nm. The second and

third mode in the distribution are far more sensitive to wind direction, with the southerly traffic-

dominated wind directions showing modes at around 32 and 76 nm as opposed to 56 nm and 263

nm for the northerly mode data. The former values compare well with modes in the number

distribution of around 20 nm and 50 nm previously attributed to the nucleation mode and Aitken

mode particles respectively from engine exhaust when sampled at Marylebone Road, with data

analysed by Positive Matrix Factorization (Harrison et al., 2011).

The Westminster University sampling site is 26 m higher and slightly displaced (~8 m) horizontally

from the Marylebone Road air sampling station. The observations at roof level are influenced by

the flow separation over the roof, if the air is entering or exiting the canyon, and the background

concentrations. The particle size data (Table S1) indicate a nucleation mode very similar in size to

that observed within the street canyon at the Marylebone Road site. Concentrations are elevated for

the 135 and 180º wind bearings suggesting that enhanced concentrations occurring within the

canyon on southerly winds are also elevated at the Westminster University sampler but the dataset

is very small and hence not included in Figure 4. The second mode appears to be broadly similar in

size to that at Marylebone Road and falls within the range of modal diameters measured at

Marylebone Road. Similarly, the third mode falls within the rather variable range also seen at

Marylebone Road.

The North Kensington site is widely taken as representative of the background air pollution climate

in central London (Bigi and Harrison, 2010; Bohnenstengel et al., 2015). At this site, the size of the

first mode in the size distributions is remarkably constant at 22-26 nm which is slightly larger than

that observed at Marylebone Road. The second mode is also less variable than at most other sites

and broadly within the range of the second mode sizes at Marylebone Road (see Table S1). The

third mode is highly variable in size with wind direction but again broadly comparable to the data



from Marylebone Road.  The Beddows et al. (2015) Positive Matrix Factorization of particle
number size distributions data from this site identified four factors contributing to the particle
number size distributions: a secondary component accounting for 4.4% of particle number with a
mode at around 250 nm, an urban background factor (43% of particle number) peaking at around 50
nm, a traffic component (44.8% of particle number) peaking at around 30 nm and a regional
nucleation component (7.8% of particle number) peaking at 20 nm.  The regional nucleation
component showed a strong seasonality with greatest prevalence in the summer months and is
thought unlikely to have contributed significantly during the period of this campaign.
Consequently, the first mode observed in our current study is very comparable to the traffic mode
observed by Beddows et al. (2015), and the second mode corresponds strongly to the urban
background factor identified by Beddows et al. (2015) who associated this factor with aged traffic
emissions and wood smoke, the latter of which is unlikely to have influenced the size distribution at
Marylebone Road significantly.

**3.2      Particle Shrinkage**
Previous London work has shown the tendency of nucleation mode traffic-generated particles
sampled within Regent's Park to have shrunk by evaporation at rates of on average 0.13 nm s$^{-1}$
(Harrison et al., 2016) while particles in the regional atmosphere typically undergo condensational
growth at a rate of about 0.6-0.9 nm h$^{-1}$ (Beddows et al., 2014).  This reflects an initial local rapid
loss of more volatile hydrocarbons, followed by a subsequent slower condensation of low volatility
species formed by atmospheric oxidation in the regional atmosphere.

Under southerly flows the Regent's University site is downwind of Marylebone Road (Fig. 1).  The
modal diameters measured at Regent's University in the nucleation mode (Table S1) are clearly
indicative of a shrinkage of particle diameter for the wind sectors 180º, 225º and 270º,
corresponding to air having passed over Marylebone Road.  These data show that the nucleation



mode is shrinking from a diameter in the range of 21-24 nm at Marylebone Road, and 22-24 nm at
Westminster University to a diameter of 14, 9 or 12 nm at the Regent's University site.  In this case,
particle shrinkage seems to be limited to those three wind sectors, with possibly some shrinkage in
the 45º wind sector, but particles in other wind sectors retain broadly similar diameters to those
measured at Marylebone Road and Westminster University.  The second particle mode and third
particle mode (where identifiable) at Regent's University are broadly similar and considerably
larger than those measured at Marylebone Road or in the limited dataset at Westminster University.

In our earlier studies of the evolution of particle sizes between Marylebone Road and Regent's Park
(Harrison et al., 2016), the nucleation mode in the Marylebone Road size distributions lay between
20-24 nm (i.e. very similar to this study).  In Regent's Park this had reduced to within the range of
6-11 nm with the largest sizes measured in the 0º wind sector and the smallest in the 180º wind
sector.  The current data show a similar general pattern, although the extent of size reduction is
smaller.  The travel distance to the Regent's University site is shorter, hence accounting in part for
less shrinkage, but the overall shrinkage rate in the current study (0.04 nm s$^{-1}$) was smaller than
previously (0.13 nm s$^{-1}$) (Harrison et al, 2016).  This is probably explained by two factors.  Firstly,
with warmer mean air temperatures (12-18ºC) evaporation would be enhanced, and secondly, as the
site used for collection of the data described in the Harrison et al. (2016) study was in the centre of
the park and further from any major highways than the Regent's University site, it may have
experienced lower vapour concentrations.  Consequently, the two datasets appear highly consistent
with one another.

Previous BT Tower site observations have reported  loss of < 20 nm particles (Dall'Osto et al.
2011). This loss  was greatest when atmospheric turbulence levels were lowest and hence the time
for ground to sampling height (160 m) transport greatest.  That analysis is not repeated in this study.
However, the nucleation mode size (Table S1) has grown slightly from the sizes measured at



Marylebone Road for the nucleation mode. It is notable that unlike the earlier results, the amplitude
of this mode at the BT Tower was substantial and slightly larger than that observed at the ground-
level background North Kensington site suggesting that there was generally good coupling between
ground-level and the Tower site.  It is notable that the first mode diameter with greatest amplitude
was for the 270º sector (Figure 4d);  this is discussed later. The particle size distribution associated
with the 225º wind sector had only one mode at 40 nm suggestive of the second solid particle mode
with complete evaporation of the semi-volatile nucleation mode.

Earlier studies have shown that particle number concentrations (< 100 nm) in a street canyon
(Olivares et al., 2007) and urban air (Hussein et al., 2006) increase with reducing temperature.  This
is consistent with the semi-volatility of nucleation mode particles from road traffic (Harrison et al.,
2016), and consequently it would be expected that the particle size distribution as well as the
number concentration would be affected by ambient temperature.  To investigate this, the size
distributions collected in the lowest quartile of air temperatures (1.1 to 3.8ºC) were compared with
those in the highest quartile of temperature (9.1 to 11.8ºC).  This showed generally higher
concentrations associated with the higher temperatures, and a clearer nucleation mode at higher
temperatures, at all sites, and most notably at Marylebone Road.  Such behaviour is contrary to
expectations, as greater evaporative losses would be expected at higher temperatures, reducing the
magnitude of the plot, or shifting the mode to smaller sizes.  To understand this effect more clearly,
wind directions with the coldest and hottest quartiles of temperature are analysed.  The coldest
periods all occurred during northerly flows (270 to 90º) and  >85% of highest quartile of
temperatures occur during southerlies (90 to 270º).  The behaviour, especially at Marylebone Road
and Regent's University therefore appears to be determined predominantly by synoptic wind
conditions.  For Marylebone Road, the street canyon flow (Figure 6) is the dominant influence and
at Regent's University the traffic sources are most proximate with southerly flows.



### 3.3    Particle Number Concentration (CPC) Data


Average diurnal variations of total particle number count derived from the Condensation Particle
Counters produced using the Openair Software Package (Carslaw and Ropkins, 2012) appear in
Figure S1.  Perhaps surprisingly, at both Marylebone Road and Westminster University, these show
a peak occurs between midnight and 6 am before reducing and then rising to a second peak in the
afternoon.  CPC concentrations at these sites far exceed those at Regent's University and the BT
Tower, whereas integrated counts from the SMPS instruments were considerably smaller and
showed a diurnal variation broadly similar to that expected for road traffic emissions (Figure 3).
While it is quite normal for the CPC to give a higher count than the SMPS since it measures over a
wider size range and may have lower internal losses (although the SMPS data analysis software
corrects for internal losses), the ratio of CPC to SMPS is typically around two, but this value was
significantly exceeded episodically, especially at Westminster University (Figure S2).  The overall
pattern of CPC to SMPS ratios (Figure 7) shows that some of the highest ratios were at Regent's
University with two individual occasions exceeding 13.  Some high peak values were observed at
Westminster University during the short SMPS time series.  Wood burning is recognised as an
influential source of particles in London (Harrison et al, 2012b; Crilley et al., 2015), and has a
diurnal profile with higher concentrations typically at night.  During the ClearfLo winter campaign
the BT Tower was influenced substantially by wood smoke  irrespective of boundary layer depth
(Crilley et al, 2015).  Since the BT Tower site was predominantly within the mixed layer during the
2017 campaign (Figure S10) and the CPC/SMPS average ratios at the Tower show little nocturnal
elevation, we consider it unlikely that wood smoke explains our observations.  Furthermore, particle
size distributions associated with biomass burning are typically larger than those from road traffic,
and outside of the sub-15 nm size range (Vu et al., 2015a).

To evaluate this phenomenon more closely, the Black Carbon data were examined.  These are
typically taken as a good tracer of diesel exhaust which is expected to be the main source of the



particle number count.  The diurnal variation in Black Carbon (Figure S3) conformed reasonably
well to that expected for a traffic-generated pollutant with Marylebone Road concentrations far
exceeding those at the other sites and showing a typical traffic-associated pattern.  Particle number
(derived from the CPC) to Black Carbon ratio (Figure S4) shows huge diurnal variability similar to
that seen in the ratio of particle number count from the CPC to that derived from the SMPS.  We
infer from this behaviour that a large number of particles smaller than the lower limit of the SMPS
and above the lower limit of the CPC (i.e. 2.5-14.9 nm for the 3776 instrument at Westminster
University and Regent's University; 4-14.9 nm for 3775 instrument at BT Tower; and 3-16.55 nm
for 3025 instrument at Marylebone Road) were present in the atmosphere.  Both the mean ratio of
CPC to SMPS (Figure S2) and CPC to Black Carbon (Figure S5) have ratios that are greatest in the
early morning (midnight to 6 am).  This is unexpected for the CPC/SMPS ratio, as the contribution
of traffic relative to regional aerosol is expected to be least and the coarser regional aerosol contains
few particles in the size range below the lower limit of the SMPS instrument.  Similarly, for the
Black Carbon data, one would expect that if traffic is the main source of particles measured by the
CPC, the latter would show a diurnal fluctuation like that of Black Carbon, which in London arises
mostly from traffic emissions.  Consequently, it seems likely that nucleation processes favoured by
the cooler temperatures and lower condensation sink in the early hours of the morning are creating
large numbers of particles in the range of 2.5-15 nm mobility diameter.  These are forming as air
moves away from the traffic source and hence are greatest at the rooftop Westminster University
site and have diminished to some extent by coagulation or re-evaporation by the time they reach the
Regent's University site which still shows a marked elevation in particle number to Black
Carbon ratio in the earlier hours of the morning compared to the Marylebone Road site.

Such behaviour is somewhat unexpected and a review of papers in which vertical gradients in
particle number count have been measured above roadside sites showed no earlier evidence of such
behaviour (Lingard et al., 2006;  Agus et al., 2007;  Nikolova et al., 2011;  Ketzel et al., 2003;



Longley et al., 2003; Kumar et al., 2008a,b; Kumar et al., 2009; Li et al., 2007; Vakeva et al.,
1999; Zhu et al., 2002; Wehner et al., 2002). However, evidence is seen in some of Villa et al.'s
(2017) observations, particle number count increased with height up to around 10 m above a multi-
lane highway. The authors reported this unexpected pattern for some ascents/descents and
attributed it to exhaust tubes of heavy duty trucks tending to project vertically upwards and to be
located at a height of several metres above ground. They suggest this is not the case in urban
canyons.

Another possibility arises from the report of Rönkkö et al. (2017) that large numbers of sub-4 nm
particles are observed in the exhaust of some diesel engines and the observation by Nosko et al.
(2017) of substantial numbers of similarly sized particles amongst emissions from brake wear.
Kontkanan et al. (2017) reported observations of sub-3 nm particles from many sites, the highest
concentrations being in urban locations. The diurnal and regional variations did not relate clearly to
photochemistry and it was concluded that sub-3 nm particle concentrations are affected by
anthropogenic sources of precursor vapours. The correlation of sub-3 nm particle concentrations in
Helsinki with nitrogen oxides suggested a link with traffic emissions. Shi et al. (2001) measured
particles of >9.5 nm by SMPS, >7 nm by CPC and >3 nm by ultrafine CPC, finding large numbers
of particles in urban air in the ranges 3-7 nm and 3-9.5 nm by differences of counts. Ratios of CPC
(>3 nm):SMPS (>9.5 nm) were highly variable, but typically around 4. Clear links to road traffic
were seen, with drive-by experiments showing large numbers of particles in the 3-7 nm range in the
exhausts of both diesel and gasoline vehicles (Shi et al., 2001). Nanoparticles were also produced
in the plume downwind of a stationary combustion source (Shi et al., 2001). Herner et al. (2011)
measured the size distribution of particles emitted from vehicles equipped with diesel particle
filters, and with diesel filters and selective catalytic reduction. The dominant mode in the size
distribution was at 10 nm diameter and comprised particles with a high fraction of sulphate. In
highway and roadside measurements in Helsinki, Enroth et al. (2016) measured particle size



distributions with a dominant mode at 10 nm diameter.  Such particles would be largely below the
lower threshold for counting by the SMPS but not the CPC.  It is plausible that during the cooler
hours of the night a tail of <2.5 nm particles might be subject to condensational growth if the co-
emitted vapour were to be supersaturated in the atmosphere within the street canyon.  The
dominance of a 10 nm mode in the size distribution would appear to be the most plausible
explanation for the high number concentration of particles observed at the Westminster University
rooftop location and the apparent transport of a substantial proportion of such particles to the
Regent's University measurement site.  While this can explain the typically high CPC/SMPS ratios
observed, it does not explain their diurnal variation.  This appears to require growth of sub-2.5 nm
particles into the range measured by CPC in the cooler, more humid nocturnal conditions.  Rönkkö
et al. (2006) and Schneider et al. (2005) studied the formation of mechanisms and composition of
diesel exhaust nucleation particles in the laboratory and during car chasing.  They conclude that
formation of nucleation mode particles depends upon formation of sulphate nuclei upon which
hydrocarbons condense, consistent with earlier studies of Shi and Harrison (1999) and Shi et al.
(2000) conducted in our laboratory.  Factors favouring nucleation mode particle formation were
found to be low temperature and high humidities, consistent with field measurements made on
Marylebone Road (Charron and Harrison, 2003).  Both factors prevail at nighttime, probably
contributing to the relative increase in 2.5–15 nm diameter particles seen most notably between
midnight and 6am (Figure S2).  Salimi et al. (2017) reported nocturnal new particle formation
events in Brisbane, Australia, finding that air masses associated with nocturnal events were
typically transported over the ocean before reaching their sampling site, but the relevance to our
study is unclear, although the maritime air might sometimes be expected to show lower temperature
and higher humidity than that from the land.

Support for our observations also comes from the very detailed measurement and modelling study
of Choi and Paulson (2016).  Measuring particle number size distribution downwind of a major



highway, they found a positive anomaly in particle number within the first 60 m of the plume peak,
as the peak for the small particles appeared further downwind than the peak in accumulation mode
particles. They attributed this to growth of unmeasured sub-5.6 nm particles into the smallest
measurable size range and suggested condensational growth or self-coagulation as the mechanism
(Choi and Paulson, 2016). Kerminen et al. (2007) measuring near a major road in Helsinki reported
particle growth by condensation to be a dominant process during the road-to-ambient evolution
stage at nighttime in winter. They inferred that under such conditions (low wind speeds with a
temperature inversion), traffic-generated particle numbers were enhanced and could affect
submicron particle number concentrations over large areas around major roads. The distance scales
for such processes in both studies (Choi and Paulson, 2016; Kerminen et al., 2007) were within 100
m of source under the conditions of measurement but might conceivably extend over greater
distance scales. Similar processes of particle evolution within an aircraft exhaust plume have been
reported by Timko et al. (2013).

Pushpawela et al. (2018) report a phenomenon of hygroscopic particle growth at nighttime, which
can potentially be mistaken for new particle formation.  This phenomenon was observed between
0.5-5.0 hours after sunset, peaking at 3.5 hours (Pushpawela et al., 2018).  This would not appear to
explain our observations, where the peak in N/SMPS and N/BC plots (Figures S2 and S5) is
greatest at 3-4 am local time, which in London in winter is some 10-11 hours after sunset.
Additionally, such a phenomenon would be expected to be unrelated to local traffic emissions, and
hence more uniform across the various sites.

**3.5      Spatial Distribution of Particles – Horizontal and Vertical**
Figure 2 shows the time series of particle concentrations from the SMPS instruments throughout the
campaign. Clearly, as expected, the Marylebone Road site shows the highest concentrations through
the campaign period due to its proximity to the road traffic source. The other sites tend to track one



another quite closely with no consistent ranking of concentrations. There are periods such as
February 1st to 3rd when Regent's University well exceeds North Kensington, but at other times,
they are very similar (e.g. 10 – 12 February), or periods when North Kensington exceeds Regent's
University (e.g. 7 February) but these are few. In the former period (1 – 3 February), winds were
southerly and concentrations at Regent's University would be enhanced by passage of air across
Central London, including Marylebone Road. In the situation where concentrations were similar (10
– 12 February), winds were in the northerly sector, giving relatively low concentrations at all sites,
and rather little spatial variation. The temporal pattern at all sites showed substantial similarity
overall (Figure 2), including diurnal patterns (Figure 3), although the magnitude of concentrations
varied.

A time series of CPC particle number concentrations (Figure 8) showed that under most conditions ,
the number count was lowest at the BT Tower site, and that the number count at Westminster
University frequently exceeded that at Marylebone Road, with Regent's University lower, but
above the concentration at the BT Tower (Figure 8). During the period of northerly winds (8 – 12
February), all sites showed low concentrations with Regent's University and BT Tower similar for
much of the time, as for the SMPS data (Figure 2). The highest CPC count concentrations during
the latter were measured at Westminster University (Figure 9) which was downwind of Marylebone
Road at those times.  The similarity seen between Westminster University and Marylebone Road
for much of the campaign, with concentrations far in excess of those at BT Tower is strongly
suggestive of continuing particle growth into the size range 2.5–14.9 nm at Westminster University
with re-evaporation occurring before reaching the elevated BT Tower site, as previously observed
by Dall'Osto et al. (2011).  Elevations in N/BC data were seen at the BT Tower site (Figure S4 and
S5) but these occurred mainly during the morning rush hour period, presumably due to fresh traffic
emissions, rather than overnight as at the other sites (Figure S5).



Figure 2 suggests that vertical gradients between the proximate Regent's University and BT Tower
sites were small in SMPS count (Figure 2), but at certain times were substantial in the CPC count
(Figure 9).  The particle size distributions measured at the BT Tower (Figure 4d) differ from
Marylebone Road and North Kensington (Figure 4a and b) in having no obvious mode in the
nucleation size range at 20 – 30 nm, a feature shared with Regent's University (Figure 4c).  Only
during westerly winds (270º) does the BT Tower show such a mode (Figure 4d), while at Regent's
University (Figure 5) the 270º wind direction also shows differences from the others with a mode at
below 20 nm. Anomalous behaviour in this wind sector is also observed at North Kensington
(Figure 4b), and at Marylebone Road. The most pronounced nucleation mode peak is associated
with the 270º and 225º wind directions.  In the Marylebone Road case, these wind directions are
almost parallel to the highway, which might explain the high concentrations and pronounced
nucleation mode, but this explanation does not work for the other sites. A more likely explanation is
that all sites are affected by emissions from Heathrow Airport which is to the west of London and
has been recognised as a major source of nucleation mode particles associated with aircraft and road
traffic emissions (Masiol et al., 2017). At a site 1 km from the northern boundary of Heathrow
Airport, PMF factors attributed to aircraft (mode at <20 nm) and fresh road traffic emissions (mode
at 18–35 nm) accounted respectively for 31.6% and 27.9% of particle number count in the warm
season and 33.1% and 35.2% in the cold season (December 2014 – January 2015) data (Masiol et
al., 2017).  Heathrow Airport is located approximately 22 km from our Central London sites on a
bearing of 255º.  Keuken et al. (2015) measured a large elevation in concentrations of particles of
10–20 nm diameter attributed to aircraft emissions (emission studies are reviewed by Masiol and
Harrison, 2014) at a site 7 km east of Schiphol Airport (Netherlands) and have shown by modelling
and measurement that concentrations are elevated to considerably greater downwind distances.
Similarly, Hudda et al. (2014) reported PNC to have increased 4 to 5 fold at 8 – 10 km downwind
of Los Angeles International Airport (USA).





The size distributions have also been analysed according with mixed layer height (MLH),
determined by ceilometer (Kotthaus and Grimmond, 2018). Both Marylebone Road (Figure S6) and
Regent's University (Figure S7) have the highest concentrations associated with the deepest MLH
class (>1000 m). This seems likely to be due to an association with southerly winds and the street
canyon circulation. Whereas, North Kensington (Figure S8) has the highest concentrations during
shallow MLH (< 100 m and 100 – 200 m) when dispersion is limited for the low altitude emissions.
The most interesting behaviour is seen at the elevated (160 m) BT Tower site, which is consistent
with Harrison et al. (2012a) and Dall'Osto et al. (2011). During the shallowest MLH (< 100 m) the
measurement site is above the inversion and the size distribution lacks an obvious nucleation mode
(Figure S9). As the MLH deepens, a nucleation mode appears which dominates the size
distribution for the deepest MLH categories (900 – 1000 m and >1000 m) with a mode at 20 – 30
nm, similar to that seen at Marylebone Road for the same MLH depths (Figure S6). The gradual
transitioning of size distribution as the MLH deepens is consistent with the surface source (mainly
road traffic) of nucleation mode particles, and their evaporative loss which increases with the
timescale of vertical mixing to the height of the sampler, as reported by Dall'Osto et al. (2011), and
the ultimate isolation of the sampler from ground-level emissions at the shallowest boundary layer
heights, as observed by Harrison et al. (2012a).

**3.6    Detailed Comparison of Marylebone Road, Westminster University and Regent's**

537       **University**

Unfortunately, a full dataset for the Westminster University site was only collected over the period
January 30th to February 1st due to a late set-up of the instrument and a malfunction after February
1st. This period however merits closer examination as it is the only period where SMPS data were
available for all three sites. For much of the time the SMPS data for the Westminster University
site looks surprisingly similar to that of the Marylebone Road site despite the former being on the
rooftop and the latter being within the street canyon. A detailed analysis hour by hour showed that



out of 51 hourly observations, in 23 the amplitude of the mode (dN/dlogD) at Westminster
University was within ± 20% of that at Marylebone Road while in 25 cases the amplitude was
greater at Westminster University than at Marylebone Road, and in just two cases the amplitude
was smaller at Westminster University.  In an attempt to explain this observation, the
meteorological data for the periods of similar magnitude and of different magnitudes were
compared but no systematic difference was seen in wind direction, air temperature or relative
humidity between any of the periods.  Wind directions were generally in a south-easterly to easterly
sector, mean temperatures around 8ºC and relative humidity high (85 and 99%). The maximum
MLH were low and there was a lot of rain (Figure  S10).

In order to gain further insight, the time series of observations were plotted for this period and
appear in Figure 9.  The SMPS integrated number counts shown in Figure 9(a) show a remarkable
similarity between Marylebone Road, Westminster University and Regent's University.  For the
first two days, Regent's University concentrations are lower than those from the other two sites,
although on the third day they are very similar to those at Westminster University.  On the first and
last days, the peak concentrations at Marylebone Road exceed those at Westminster University but
on the middle day (January 31$^{st}$) the differences between these two sites are very small.  The CPC
particle number counts shown in Figure 9(b) are very similar to those at Marylebone Road on the
first and last day but exceed those at Marylebone Road on January 31$^{st}$.  Concentrations at Regent's
University are typically only around half or less of those measured at Westminster University.  The
magnitude of the CPC concentrations peaking at over 40,000 cm$^{-3}$ is close to double the integrated
SMPS counts which peak at a little over 20,000 cm$^{-3}$ indicating a large number of particles in the
size range below 14.9 nm.

However, the Black Carbon data (Figure 9c) have daytime concentrations at Marylebone Road that
far exceed those at Westminster University and Regent's University, the latter sites tracking each





other and having very similar concentrations.  Since Black Carbon can be viewed as a conserved
tracer of vehicle emissions over these small time and distance scales, the inference is that particle
production must be continuing as the vehicle exhaust mixes upwards from the street canyon
Marylebone Road site to the Westminster University rooftop site.  The southerly wind directions
likely associated with upward flow on the Westminster University canyon wall (Fig. 6) would carry
vehicle exhaust past the Marylebone Road measurement station (south side of the road).

Air leaving the canyon and being entrained by the complex building roof flows could expose the
Westminster University sampler to air exiting the street canyon and to the general flow towards
Regent's University site  (Fig. 6 and 1).  Such behaviour is consistent with the observations of
particle growth in the sub-SMPS size ranges reported in the previous section extending into the
SMPS size range.  This is similar to behaviour observed by Kerminen et al. (2007) in Helsinki who
observed not only possibly evaporation of some particles in the 7–30 nm range, but also on apparent
growth of nucleation mode particles into the 30–63 nm size range between sampling points at 9 m
and 65 m downwind of a highway.  The results in Figure 9 are suggestive of a substantial growth of
nuclei into the range of the CPC at Westminster University.

**4.       CONCLUSIONS**
The measurement of particle number size distributions in the atmosphere is resource intensive and
there have been rather few studies in which more than two samplers have been operated within a
city.  Typically if there are two sites, one is a traffic-influenced site and the other urban background.
In this study, data have been collected at a total of five sites, although unfortunately the dataset
from the Westminster University site is limited to only a few days.  Nonetheless, the dataset allows
some deep insights into the spatial distribution of particle sizes and number counts not only
horizontally but in the vertical dimension.  Not unexpectedly, concentrations of particles at the
street canyon Marylebone Road site considerably exceed concentrations at other sites, but there are





nonetheless considerable similarities in diurnal profiles and the magnitude of concentrations at the
other, background sites.

One of the main motivating factors for this study was to confirm earlier observations of shrinkage
of the nucleation mode particles between traffic emissions on Marylebone Road and the downwind
site at Regent's University within Regent's Park.  Particle shrinkage was observed within the
current study although at a slower mean rate (0.04nm s$^{-1}$) than in the earlier study (Harrison et al.,
2016) in which the mean shrinkage rate was 0.13nm s$^{-1}$.  However, temperatures in the current
study all fell below those in the earlier work of Harrison et al. (2016).  Other factors may also have
been influential.  There have been marked changes in the road vehicle fleet in London between the
two measurement campaigns.  The earlier dataset as reported by Dall'Osto et al. (2011) and
Harrison et al. (2016) was collected in 2006 at which time the sulphur content of diesel fuel was
regulated at below 50 ppm.  Between the two campaigns, the sulphur content of both gasoline and
diesel motor fuels was reduced to below 10 ppm sulphur in order to facilitate the introduction of
diesel particle filters from 2011 onwards.  The incorporation of a diesel particle filter on EURO 5
and EURO 6 vehicles leads to a substantial overall reduction in particulate matter emissions but
also a change in the hydrocarbon content of the particles.  Secondly, the Regent's Park sampling
site used for the 2006 measurements was at about double the distance from Marylebone Road
compared to the Regent's University used in the latest study.  This would allow for greater dilution
of the traffic plume from Marylebone Road and other adjacent highways, leading to a greater
reduction in vapour phase hydrocarbons at the more distant site causing an accelerated evaporation
process.  The reduction in fuel sulphur content in 2007 was accompanied by a marked change in the
size distribution of particles emitted from road traffic, including a reduction in the nucleation mode
particles (Jones et al., 2012).  The work of Dall'Osto et al. (2011) also analysed data from the BT
Tower, showing increasing evaporative loss of nucleation mode particles as the travel time from
ground level to the sampling site on the Tower became longer with reduced atmospheric turbulence





levels.  Although that phenomenon has not been studied in detail in the latest dataset, the results are
clearly consistent with such a process, and with an apparent total loss of the nucleation mode in
particles associated with regional pollution sampled when the boundary layer top was below the
sampling height on the tower.

Although the phenomenon of particle shrinkage had been seen in earlier work, there were two
further major observations made in the current study which were not anticipated.  The first, was the
clear influence of a major source to the west of London, almost certainly Heathrow Airport, upon
concentrations of nucleation mode particles.  The association of an enhanced nucleation mode in the
270° or 225° sector is indicative of a major source of very fine particles, and the work of Masiol et
al. (2017) at a sampling site close to Heathrow Airport provides strong evidence for major
emissions both from aircraft engines and the large volumes of road traffic attracted by the airport.
Earlier research by Keuken et al. (2015) and Hudda et al. (2014) gives a clear precedent for
measurement of strongly elevated concentrations of very fine particles several kilometres
downwind of a major airport, but to our knowledge this is the first observations of concentrations
above urban background at a distance of 22 km from the centre of the airport.

The other observation which was wholly unexpected was of the very poor relationship between total
particle numbers measured by the Scanning Mobility Particle Sizers and the total particle numbers
measured by co-located condensation particle counters.  While both the SMPS counts and co-
located Black Carbon measurements show a typical road traffic diurnal profile, the CPC data show
a quite different diurnal profile peaking at night.  This is most evident in the ratios of CPC/SMPS
and CPC/BC seen at all sampling sites, with the exception of CPC/BC at the elevated BT Tower
site which does not show a nocturnal maximum, but peaks during the morning rush hour period.
Earlier studies such as that of Choi and Paulson (2016) and Kerminen et al. (2007) have reported
data consistent with such a phenomenon, but with very modest elevations in particle count




compared to those in the current data.  The implication is of the presence of large numbers of
particles within the range of 2.5 – 15nm and hence observable with the CPC but below the lower
cut of the SMPS.  It seems likely that such particles grow at night from vary small nuclei and it
seems possible that the exceptional magnitude of this process within London results from the high
density of diesel traffic leading to substantial nocturnal concentrations of condensable vapours close
to the traffic source.  A common feature to such observations appears to be its association with still
conditions on winter nights which lead to poor dispersion of vehicle emissions and a pool of vapour
co-emitted with traffic particles which becomes supersaturated as it cools in the ambient
atmosphere, leading to condensation on small nuclei when the general particle concentrations and
hence the condensation sink are relatively low in magnitude.

These very abundant particles within the 2.5 – 15 nm range are likely to prove ephemeral as they
would be expected to re-evaporate as the air mass dilutes away from source.  However, the health
effects of exposure to particles within this range are poorly known and no recommendation can be
given as to whether health-related studies would be best to measure the particle size range covered
by the SMPS as is most typically performed at present, or whether CPC data going down to smaller
particles sizes would be more appropriate.

There are some additional general conclusions from the work.  Firstly the results demonstrate the
dynamic behaviour of traffic-generated (and other) particles within the urban atmosphere.  Our
earlier paper (Dall'Osto et al., 2011) referred to "remarkable dynamics", and further remarkable
dynamic processes have been observed in the current study.  Secondly, as this work has revealed
sources and processes that were not originally anticipated, although with the benefit of hindsight it
might have been possible to predict them, there is clearly a need for further detailed observational
studies of the behaviour of sub-100 nm particles within the urban atmosphere.



**ACKNOWLEDGEMENTS**
The authors are grateful to the management and  staff of Westminster University, Regent's
University and British Telecom for access to their buildings for air sampling.  They also express
gratitude to the National Centre for Atmospheric Science (NCAS) for the loan of sampling
instruments, and to Dr Paul Williams (NCAS) for facilitating the instrument intercomparison.  The
operation of the ceilometers were supported by NERC ClearfLo, NERC AirPro, Newton Fund/Met
Office CSSP (SG, SK) and University of Reading. We acknowledge the support of KCL LAQN for
the instrument sites and support and the Reading Urban Micromet group for maintaining the
instruments, notably in this period Elliott Warren and Kjell zum Berge. The work was funded by the
European Research Council (ERC-2012-AdG, Proposal No. 320821) and the UK Natural
Environment Research Council (R8/H12/83/011) and a NCAS studentship (to JB).





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



**TABLE LEGENDS**

**Table 1:**    Location sites of instruments during the campaign. Mean sea level (msl), Above ground level (agl), Condensation particle counter (CPC), Scanning Mobility Particle Sizers (SMPS).

**FIGURE LEGENDS**

**Figure 1:**    Study area locations (a) in central London (UK) and (b) more detail of the Marylebone Road (MR), Westminster University (WU) and Regent's University (RU) sites.

**Figure 2:**    Time series of total particle number count from the SMPS instruments at the five sites (Fig. 1, Table 1) over the campaign period.

**Figure 3:**    Campaign-average diurnal variation of particle number counts derived from the SMPS instruments with median (line) and inter-quartile range (shading) shown.

**Figure 4:**    Average particle number size distributions stratified by 45° wind directions sectors (°, measured at LHR, value indicates mid-point of sector ers) for (a) Marylebone Road, (b) North Kensington (c) Regent's University, (d) BT Tower.

**Figure 5:**    Lognormal modes fitted to the average particle size spectrum at North Kensington for wind direction sector 270°.

**Figure 6:**    A schematic diagram of the wind flows in the street canyon of Marylebone Road (6 traffic lanes) during southerly and northerly winds. The orange marker represents the MR sampling site and red marker represents the WM sampling site.

**Figure 7:**    Time series (15 min) of ratio of total particle number counts, CPC/SMPS, for four sites over the campaign period.

**Figure 8:**    Time series (15 min) of total particle number count from the CPC instruments located at four sites over the campaign period.

**Figure 9:**    Time series (15 min) of (a) SMPS integrated counts, (b) particle number counts (CPC) and (c) Black Carbon from Marylebone Road, Westminster University and Regent's University for 30 January to 1 February 2017.



**Table 1:** Location sites of instruments during the campaign. Mean sea level (msl), Above ground level (agl), Condensation particle counter (CPC), Scanning Mobility Particle Sizers (SMPS).

| Site Name | Marylebone Road | Westminster University | Regent's University | BT Tower | North Kensington |
|---|---|---|---|---|---|
| Lat (°N), Long (°W) | 51.522530, 0.154611 | 51.522322.0.15515 | 51.525542, 0154570 | 51.521426, 0.138924 | 51.521082, 0.213403 |
| Height of ground msl (m) | 26 | 26 | 30 | 25 | 23 |
| Height of inlets agl (m) | 4 | 26 | 17 | 160 | 3 |
| Instruments installed | Long_DMA_SMPS/ CPC Vaisala CL31 | Long_DMA_SMPS/CPC/ (Micro)Aethalometer/Anemometer | Long_DMA_SMPS/ Short_DMA_SMPS/ CPC/Aethalometer/Anemometer | Long_DMA_SMPS/CPC/ (Micro) Aethalometer/Anemometer | Long_DMA_SMPS Vaisala CL31 |
| Particle spectrometer type | 3080+3081+3775 | 3080+3081+3776 | (3082+3081+3775)/(3082+3085+3776) | (3080+3081+3775) | (3080+3081+3775) |
| Aerosol dryer | Yes | No | No | No | Yes |
| CPC type | TSI 3025 | TSI 3776 | TSI 3776 | TSI 3775 | None |





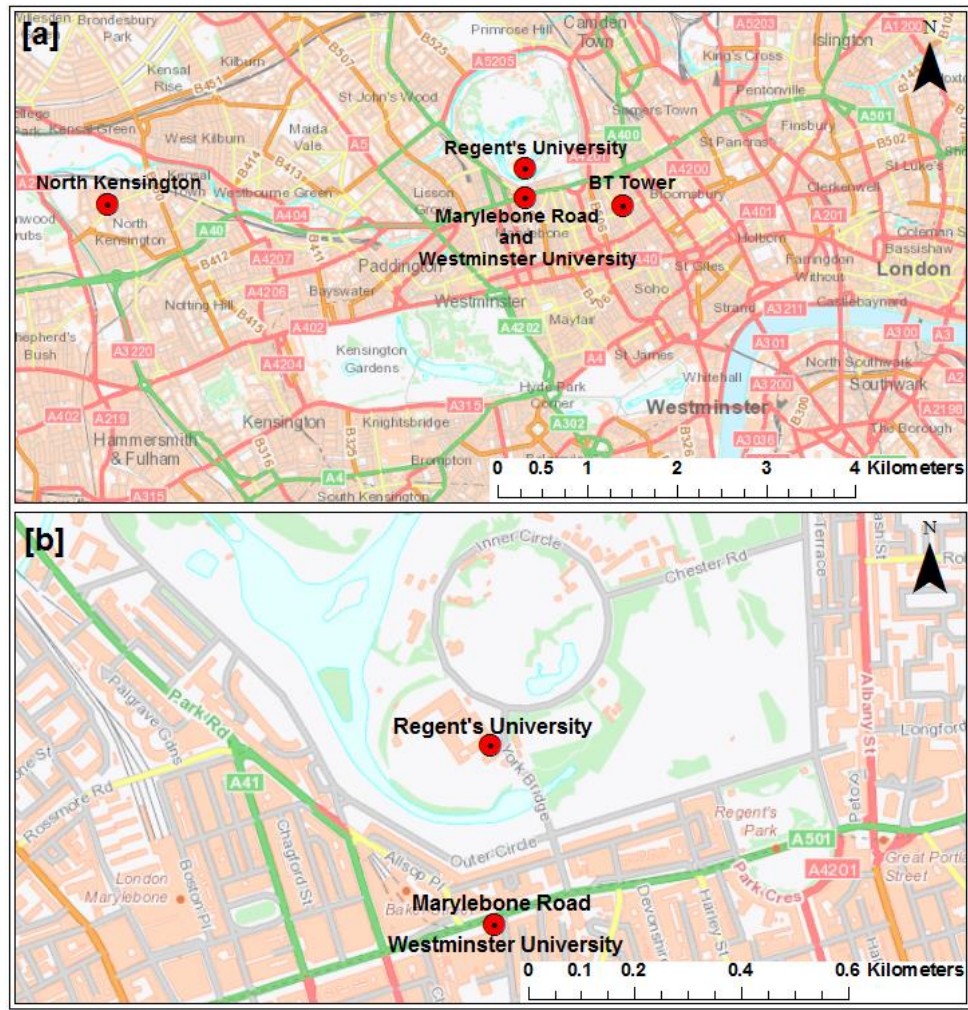


**Figure 1:** Study area locations (a) in central London (UK) and (b) more detail of the Marylebone Road (MR), Westminster University (WU) and Regent's University (RU) sites.





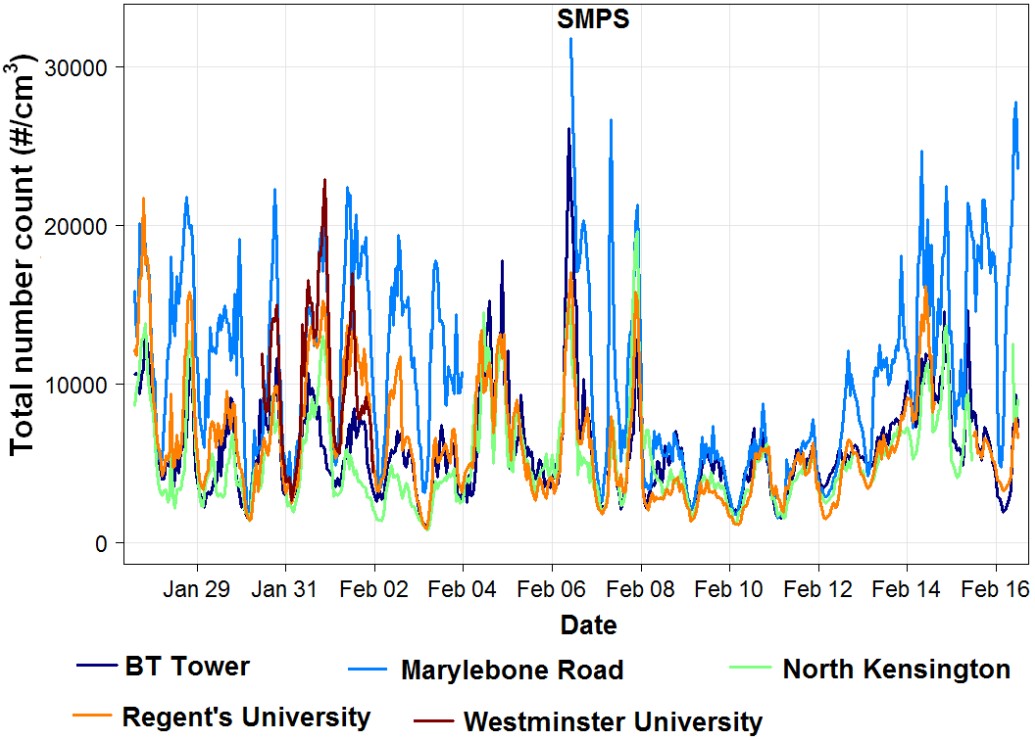



**Figure 2:** Time series of total particle number count from the SMPS instruments at the five sites
(Fig. 1, Table 1) over the campaign period.





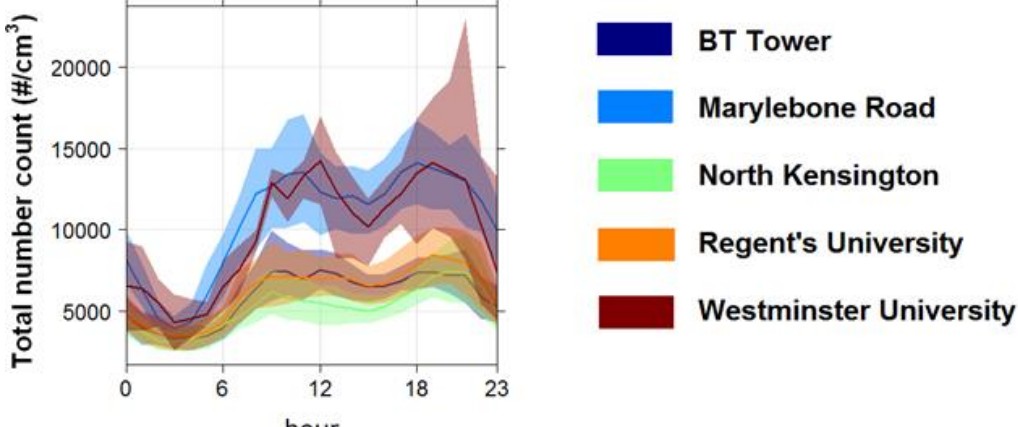

**Figure 3:** Campaign-average diurnal variation of particle number counts derived from the SMPS
instruments with median (line) and inter-quartile range (shading) shown.





**(a)**

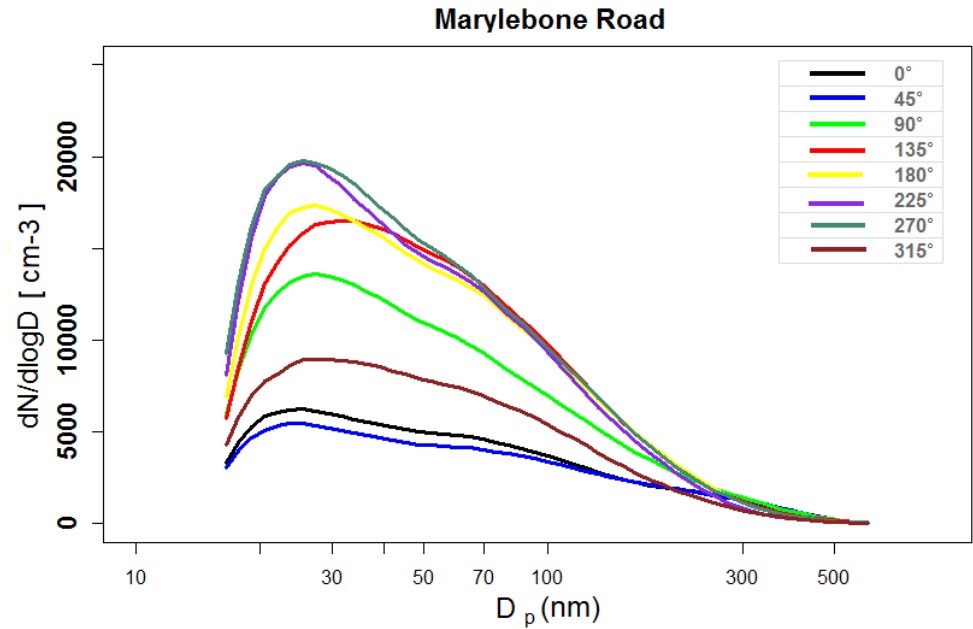


**(b)**

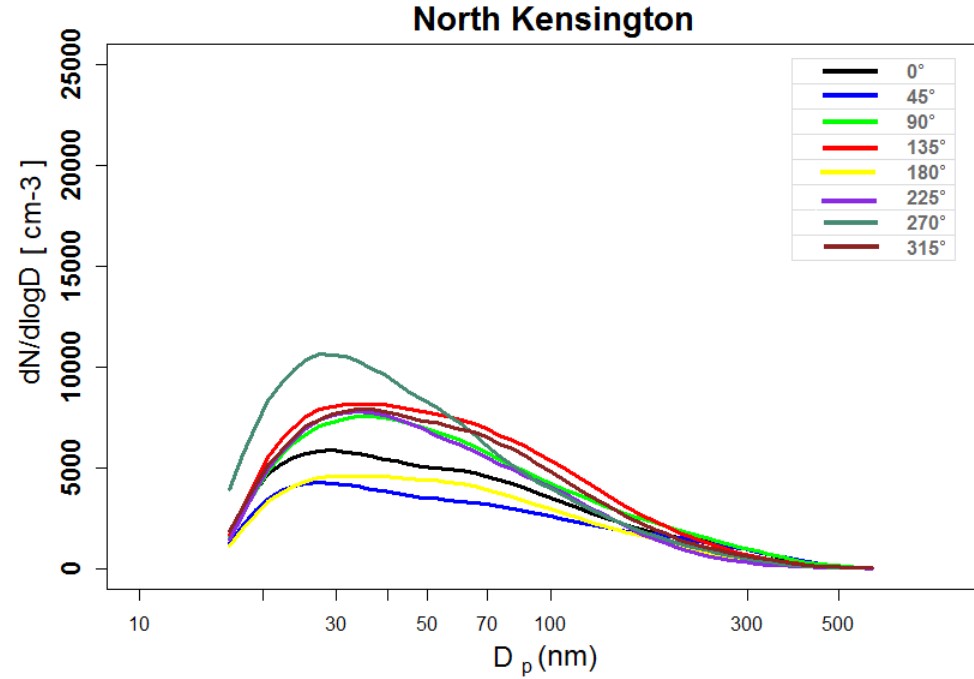







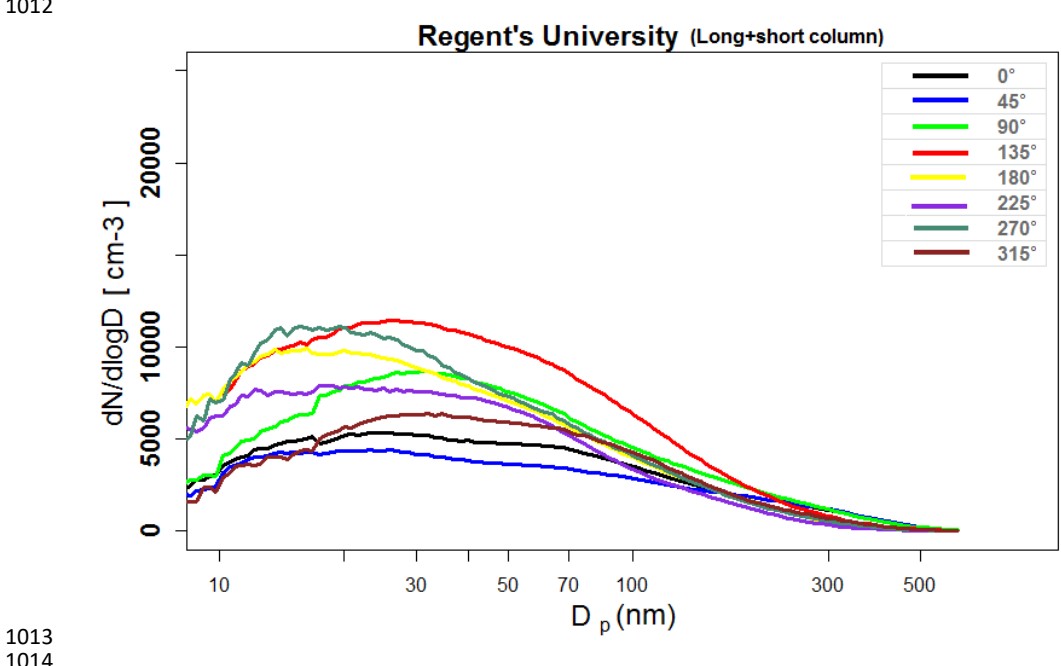


**(d)**

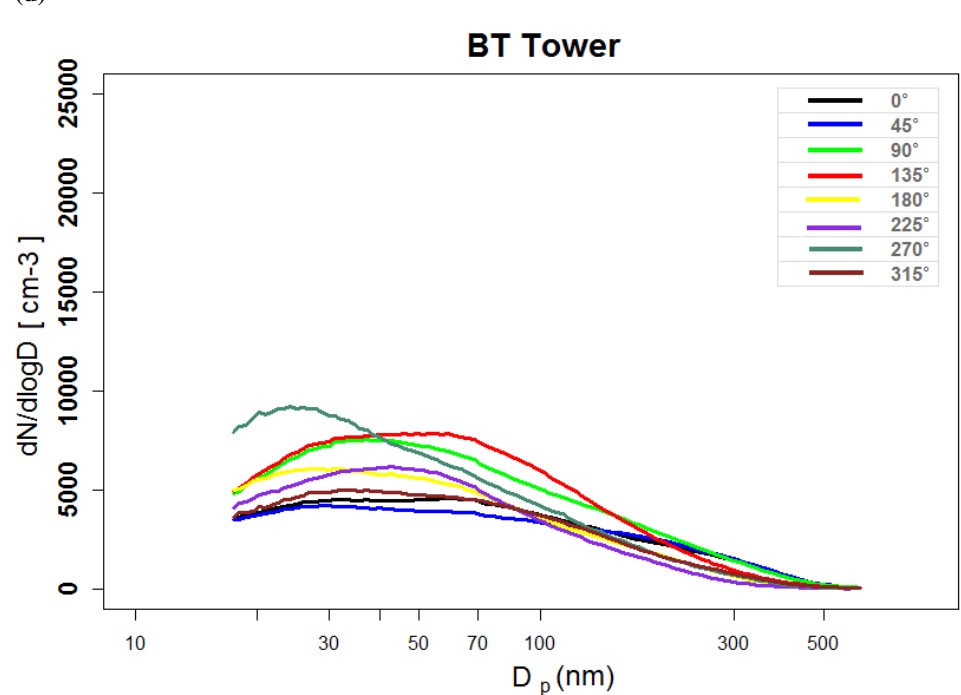

**Figure 4:** Average particle number size distributions stratified by 45° wind directions sectors (°,
measured at LHR, value indicates mid-point of sector ers) for (a) Marylebone Road, (b) North
Kensington (c) Regent's University, (d) BT Tower.






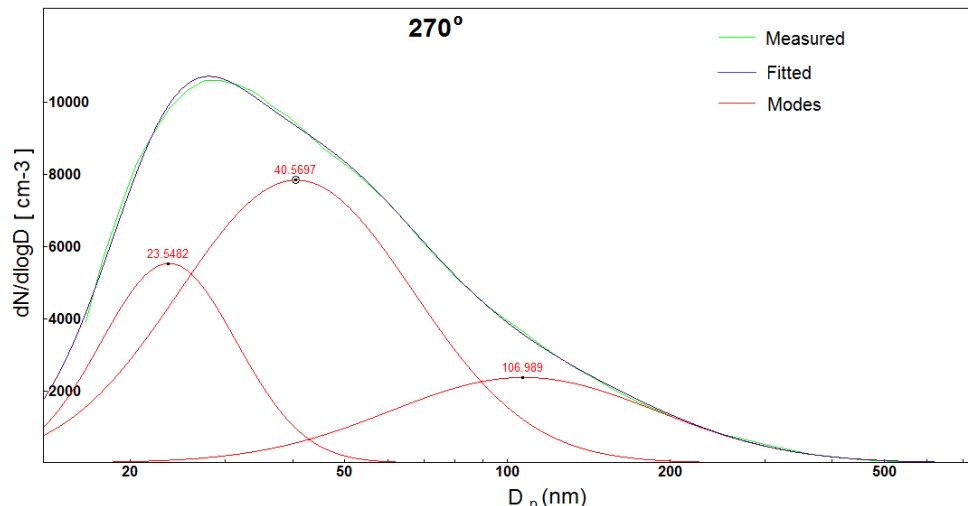


**Figure 5:** Lognormal modes fitted to the average particle size spectrum at North Kensington
for wind direction sector 270º.




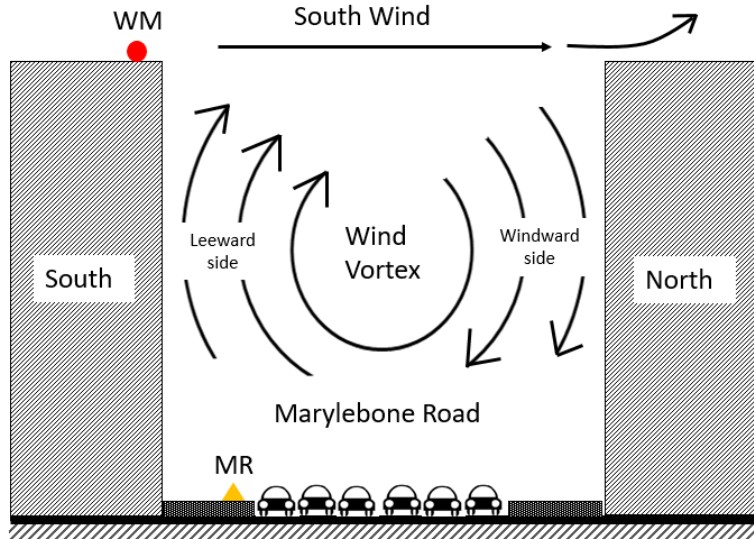


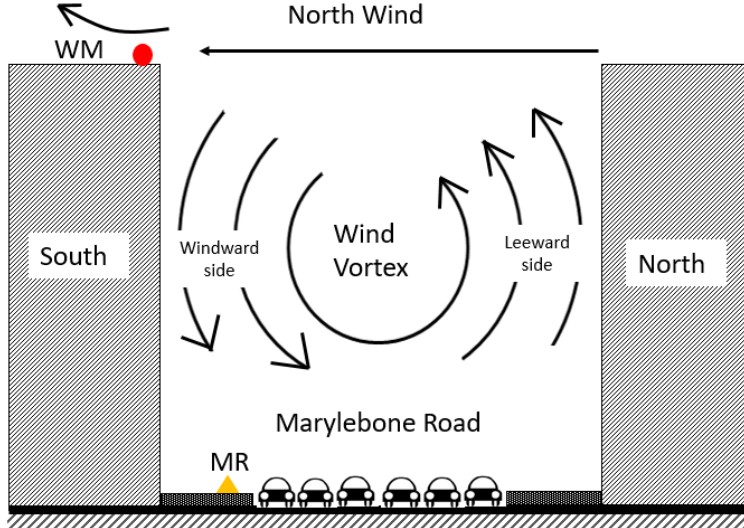


**Figure 6:** A schematic diagram of the wind flows in the street canyon of Marylebone Road (6
traffic lanes) during southerly and northerly winds. The orange marker represents the MR sampling
site and red marker represents the WM sampling site.







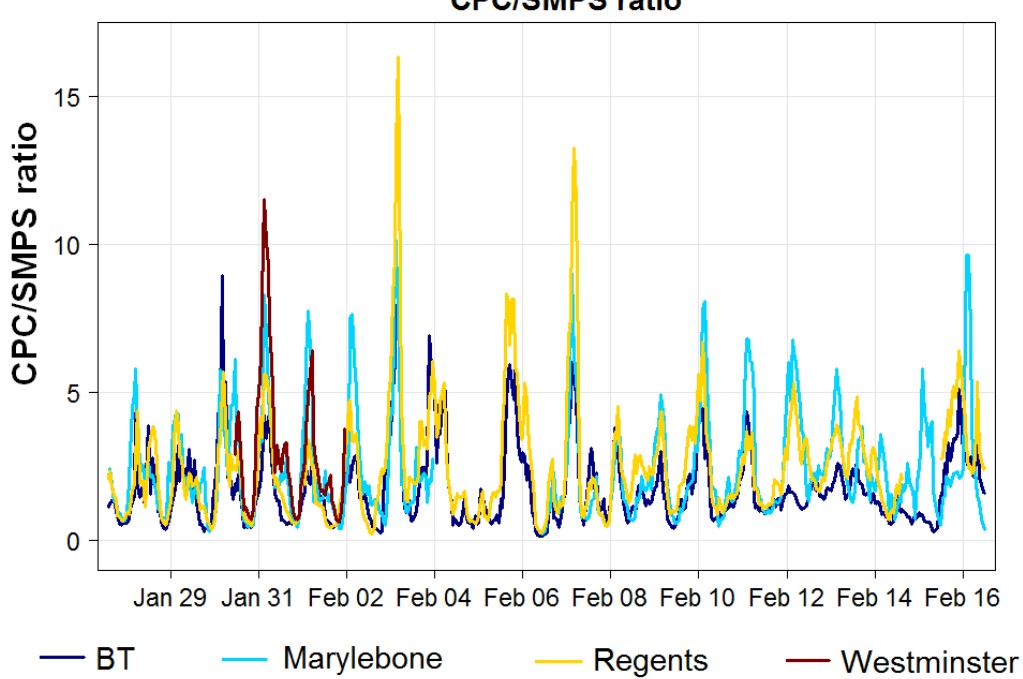

**Figure 7:** Time series (15 min) of ratio of total particle number counts, CPC/SMPS, for four sites
over the campaign period.

















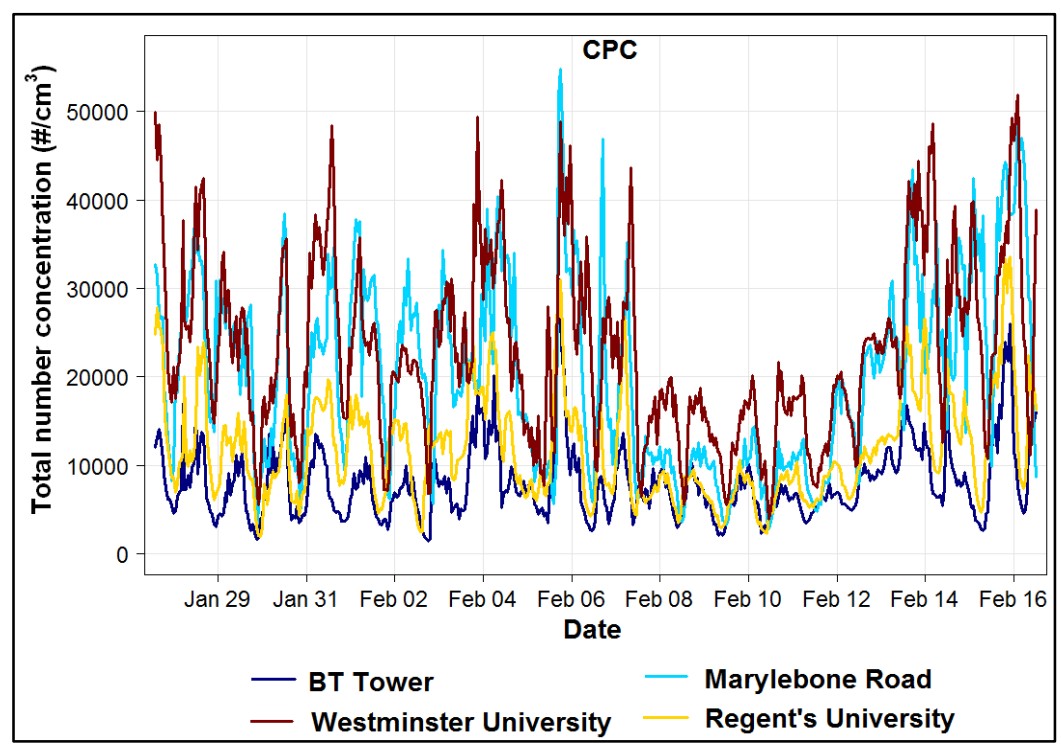


**Figure 8:** Time series (15 min) of total particle number count from the CPC instruments located at
four sites over the campaign period.





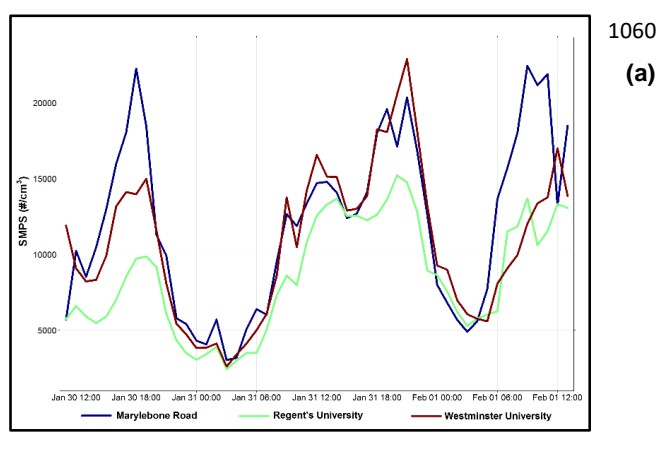

(a)


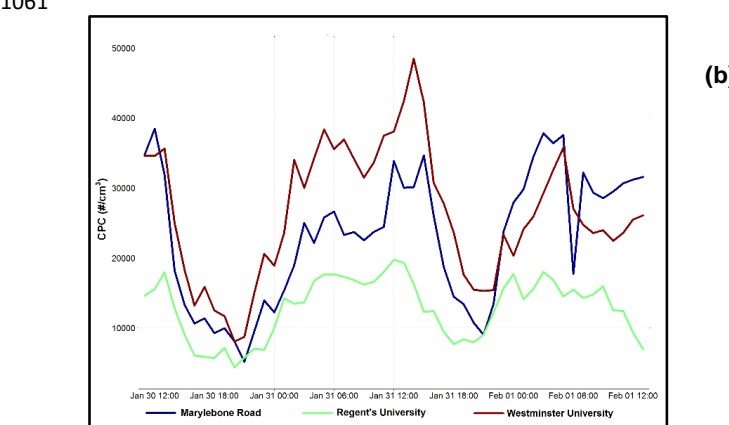

(b)



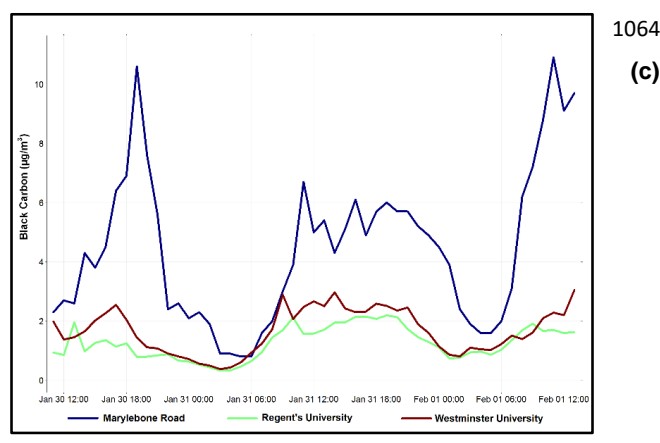

(c)

**Figure 9:** Time series (15 min) of (a) SMPS integrated counts, (b) particle number counts (CPC)
and (c) Black Carbon from Marylebone Road, Westminster University and Regent's University for
30 January to 1 February 2017.