# Peer review of "Interpretation of Particle Number Size Distributions Measured across an Urban Area during the FASTER Campaign"

_Atmospheric Chemistry and Physics, 2018_

## Referee Comment (RC1) · Anonymous Referee #1 · 24 Oct 2018

General comments

This paper gives a very thorough analysis of SMPS, CPC and aethalometers data collected at five sites in Central London during a one month campaign in winter 2017. This paper is well-written with an excellent description of scientific methods and experiments (including the limits of using different instruments – that was highly appreciated), and with appropriate amounts of supplementary material and references. This work can be seen as an important contribution to the understanding of the fate, behaviour and sources of nucleation mode particles in the urban atmosphere. Since many results are of high relevance to the community, I recommend publication in ACP after considering

a few minor comments.

Specific comments

Figure 5 shows an excellent fitting of the modes of particle size distributions corresponding to observations associated with the wind direction sector 270° at North Kensington. What about the quality of the other modal fittings (corresponding to results presented in table S1)? I would like to ask the authors to add the numbers of observations per wind sector from which modal parameters are computed (in table S1, to support results presented in Figure 4). Such information will complement usefully the analysis of results. Indeed low numbers of observations for specific wind sectors is highly possible during a short campaign corresponding to quite homogeneous meteorological conditions (here winter conditions). Low significance of data for a wind sector due to low number of observations may explain a few results that could be difficult to interpret (e.g. no particle shrinkage observed for the wind sector 135° at Regent's park, while air masses would roughly come from Marylebone Rd if I am not mistaken).

Technical corrections

It would be better for the reader if the authors could change the order of figures in the paper according to the discussion. Could you please indicate the meaning of FWHM in the supplementary material. And also indicate in part 2.3 if "LHR" (line 212 p10 ; and in supplementary material) is the abbreviation for "London Heathrow airport" (if I have well understood).

---

## Referee Comment (RC2) · Anonymous Referee #2 · 26 Oct 2018

The paper deals with a 1-month long campaign dedicated to ultrafine particles in urban environment in London. The measurements were realized in winter 2017 at 5 sites using appropiate instrumentation including SMPS and CPC among others. The work presented here has an important contribution to the urban characteristics and features of UF particles. The paper is well written, uses adequate scientific methods, and has important conclusions. Therefore, I recommend the publication in ACP after considering the following minor comments.

General comments:

1. L131: "The instruments (Table 1) were operated according to Wiedensohler et al.

[Figure]

(2012) guidelines and calibrated and intercompared both before and after the sampling campaign." There is a contradiction since dryer was not used at 3 sites and therefore how could you keep the RH below 40% as it is written in the cited paper? Also some additional data on the intercomparison would be adventageous since differrent type of CPCs were used, and the manufacturer provides 10% uncertainty between identical CPCs?

2. The work and measurements were dedicated to ultrafine particles, and nucleation mode was extensively examined in the paper. However, there is no clear evidence of atmospheric nucleation and subsequent growth in the paper. Could you provide some info regarding this?

3. In the Introduction section there are only Harrison co-authored papers cited except the authoritative reviews that altogether seems to be inadequate. Please take a deeper overview.

4. Please indicate the numbers of observations per wind sector since the representativity is not evident.

5. The street canyon wind flow diagram is highly appreciated.

6. Have you considered using median size distributions instead of average? Or is there any specific reason doing this way?

Technical comments:

L95: "Data recovery was high at all sites": Please quantify it or at least reformulate the sentence. The CPC and SMPS size ranges should be added to the Table 1 as well.

L339: Remove value judgement: "Perhaps surprisingly"

L346: "the ratio of CPC to SMPS is typically around two": Why is it typical? Please clarify it.

L457: Could that be related to other emission sources e.g. residental heating due to

its uniformity or other (meteorological) processes?

L542: Remove value judgement: "surprisingly"
* * *

---

## Author Comment (AC1) · 3 Dec 2018

RESPONSE TO REVIEWERS REFEREE #1 General comments This paper gives a very thorough analysis of SMPS, CPC and aethalometers data collected at five sites in Central London during a one month campaign in winter 2017. This paper is well-written with an excellent description of scientific methods and experiments (including the limits of using different instruments – that was highly appreciated), and with appropriate amounts of supplementary material and references. This work can be seen as an important contribution to the understanding of the fate, behaviour and sources of nucleation mode particles in the urban atmosphere. Since many results are of high

relevance to the community, I recommend publication in ACP after considering a few minor comments.

Specific comments Figure 5 shows an excellent fitting of the modes of particle size distributions corresponding to observations associated with the wind direction sector 270_ at North Kensington. What about the quality of the other modal fittings (corresponding to results presented in table S1)? I would like to ask the authors to add the numbers of observations per wind sector from which modal parameters are computed (in table S1, to support results presented in Figure 4). Such information will complement usefully the analysis of results. Indeed low numbers of observations for specific wind sectors is highly possible during a short campaign corresponding to quite homogeneous meteorological conditions (here winter conditions). Low significance of data for a wind sector due to low number of observations may explain a few results that could be difficult to interpret (e.g. no particle shrinkage observed for the wind sector $135°$ at Regent's park, while air masses would roughly come from Marylebone Rd if I am not mistaken). RESPONSE: The reviewer makes a good point. Table S1 has been revised to include the number of observations in each sector. The absence of obvious shrinkage in the $135°$ wind sector is a little surprising, although there may be mixing with the larger mode seen in the $90°$ degree data, suggestive of traffic aerosol aged over much longer periods.

Technical corrections It would be better for the reader if the authors could change the order of figures in the paper according to the discussion. Could you please indicate the meaning of FWHM in the supplementary material. And also indicate in part 2.3 if "LHR" (line 212 p10; and in supplementary material) is the abbreviation for "London Heathrow airport" (if I have well understood). RESPONSE: The sequence of figures is determined by the order in which they are cited in the text (in line with normal editorial practice). This has been checked and an amendment made to the S.I.

The term FWHM refers to Full Width at Half Maximum and is now defined in a footnote to Table S1.

[Figure]

LHR is indeed short for London Heathrow Airport, and this is now made clear in the text.

REFEREE #2 The paper deals with a 1-month long campaign dedicated to ultrafine particles in urban environment in London. The measurements were realized in winter 2017 at 5 sites using appropriate instrumentation including SMPS and CPC among others. The work presented here has an important contribution to the urban characteristics and features of UF particles. The paper is well written, uses adequate scientific methods, and has important conclusions. Therefore, I recommend the publication in ACP after considering the following minor comments.

General comments: 1. L131: "The instruments (Table 1) were operated according to Wiedensohler et al. (2012) guidelines and calibrated and intercompared both before and after the sampling campaign." There is a contradiction since dryer was not used at 3 sites and therefore how could you keep the RH below 40% as it is written in the cited paper? Also some additional data on the intercomparison would be advantageous since different type of CPCs were used, and the manufacturer provides 10% uncertainty between identical CPCs? RESPONSE: This sentence has been amended to clarify the fact that a dryer was not used at all sites. The point is discussed in some detail in the paragraph following. The CPC intercomparison, as stated in the text, revealed differences of <5% and was run for sufficiently long to generate well fitted relationships between instruments which were used to correct data to a common calibration.

2. The work and measurements were dedicated to ultrafine particles, and nucleation mode was extensively examined in the paper. However, there is no clear evidence of atmospheric nucleation and subsequent growth in the paper. Could you provide some info regarding this? RESPONSE: The campaign was conducted in winter, and no clear nucleation events were seen at any of the sites. A sentence has been added to make this clear. We have since carried out an in-depth study of a long-term dataset from three sites which include the North Kensington and Marylebone Road sites also used in this study. This can be accessed at Bousiotis, D., Dall'Osto, M., Beddows, D. C.

S., Pope, F. D. and Harrison, R. M.: Analysis of new particle formation (NPF) events at nearby rural, urban background and urban roadside sites, Atmos. Chem. Phys. Discuss., https://doi.org/10.5194/acp-2018-1057, 2018.

3. In the Introduction section there are only Harrison co-authored papers cited except the authoritative reviews that altogether seems to be inadequate. Please take a deeper overview. RESPONSE: We thank the reviewer for pointing this out. We have extended the Introduction to include further topics and references.

4. Please indicate the numbers of observations per wind sector since the representativity is not evident. RESPONSE: This is the same point as was raised by Referee #1, and the numbers of observations in each wind sector are now included in Table S1.

5. The street canyon wind flow diagram is highly appreciated.

6. Have you considered using median size distributions instead of average? Or is there any specific reason doing this way? RESPONSE: Median size distributions were not used. As the variability between size distributions for a given site and wind sector was rather small, it was felt that little would be gained.

Technical comments: L95: "Data recovery was high at all sites": Please quantify it or at least reformulate the sentence. The CPC and SMPS size ranges should be added to the Table 1 as well. RESPONSE: The sentence has been qualified for clarity. The size ranges have been added as a footnote to Table 1, with the SMPS ranges being cross-referred to the text to avoid unnecessary repetition.

L339: Remove value judgement: "Perhaps surprisingly" RESPONSE: This has been removed.

L346: "the ratio of CPC to SMPS is typically around two": Why is it typical? Please clarify it. RESPONSE: The words "in our experience" have been added with a reference as an example.

L457: Could that be related to other emission sources e.g. residential heating due to

its uniformity or other (meteorological) processes? RESPONSE: We consider other possibilities such as domestic heating by natural gas. However, the peak in N/SMPS and N/BC comes at a time (3-4am) when most domestic heating systems are shut down, or operating at a low level. A sentence has been added at the end of the first paragraph of Section 3.3 to make this point. Meteorological processes were considered and are discussed in the paper.

L542: Remove value judgement: "surprisingly" RESPONSE: The word has been deleted.